

**PeerJ Hubs**

Published on behalf of

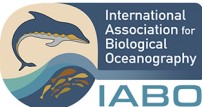

# Evaluation of DNA metabarcoding for identifying fish eggs: a case study on the West Florida Shelf

Mya Breitbart, Makenzie Kerr, Michael J. Schram, Ian Williams, Grace Koziol, Ernst Peebles and Christopher D. Stallings

College of Marine Science, University of South Florida, Saint Petersburg, Florida, United States

## ABSTRACT

A critical factor in fisheries management is the protection of spawning sites for ecologically and economically important fish species. DNA barcoding (*i.e.*, amplification and sequencing of the mitochondrial cytochrome c oxidase I (COI) gene) of fish eggs has emerged as a powerful technique for identifying spawning sites. However, DNA barcoding of individual fish eggs is time-consuming and expensive. In an attempt to reduce costs and effort for long-term fisheries monitoring programs, here we used DNA metabarcoding, in which DNA is extracted and amplified from a composited sample containing all the fish eggs collected at a given site, to identify fish eggs from 49 stations on the West Florida Shelf. A total of 37 taxa were recovered from 4,719 fish eggs. Egg distributions on the West Florida Shelf corresponded with the known habitat types occupied by these taxa, which included burrower, coastal pelagic, epipelagic, mesopelagic, demersal, deep demersal, commensal, and reef-associated taxa. Metabarcoding of fish eggs was faster and far less expensive than barcoding individual eggs; however, this method cannot provide absolute taxon proportions due to variable copy numbers of mitochondrial DNA in different taxa, different numbers of cells within eggs depending on developmental stage, and PCR amplification biases. In addition, some samples yielded sequences from more taxa than the number of eggs present, demonstrating the presence of contaminating DNA and requiring the application of a threshold proportion of sequences required for counting a taxon as present. Finally, we review the advantages and disadvantages of using metabarcoding *vs.* individual fish egg barcoding for long-term monitoring programs.

## INTRODUCTION

A critical factor in fisheries management is protecting spawning sites for ecologically and economically important fish species. Studies commonly hindcast spawning sites based on the locations where larvae from a given species have been identified, but this method is imprecise because larvae can be days, weeks, or even months old at the time of capture (*Cowen & Sponaugle, 2009*). In contrast, predicting spawning sites based on the presence of eggs is much more reliable since eggs behave as relatively passive particles (the

Corresponding author
Mya Breitbart, mya@usf.edu

exception being the eggs of species that are not neutrally buoyant) and most fish remain in this developmental stage for a maximum of 1–2 days. Additionally, the developmental stage of the eggs can be determined, allowing identification of eggs that are less than a few hours old, if so desired. However, since fish eggs are difficult to identify visually, the spawning locations of many broadcast spawning species remain unknown (*Kawakami, Aoyama & Tsukamoto, 2010*; *Becker et al., 2015*). DNA barcoding (*i.e.*, amplification and sequencing of the mitochondrial cytochrome c oxidase I (COI) gene) of individual fish eggs has emerged as a powerful technique for identification of fish spawning sites (*Shao, Chen & Wu, 2002*; *Saitoh, Uehara & Tega, 2009*; *Lelièvre et al., 2012*; *Burghart et al., 2014*; *Frantine-Silva et al., 2015*; *Harada et al., 2015*; *Lewis et al., 2016*; *Leyva-Cruz et al., 2016*; *Lin et al., 2016*; *Hofmann et al., 2017*; *Ahern et al., 2018*; *Duke, Harada & Burton, 2018*; *Burrows et al., 2019*; *Hou et al., 2020*; *Kerr et al., 2020*; *Mateos-Rivera et al., 2020*; *Hou et al., 2022*; *Lira et al., 2023*).

Through several pilot studies and the long-term Spawning Habitat & Early-life Linkages to Fisheries (SHELF) program funded by the Florida RESTORE Act Center of Excellence Program (FLRACEP), we have used DNA barcoding to identify over 8,500 individual fish eggs from over 320 locations in the Gulf of Mexico (GOM) and Florida Straits in the past decade (*Burghart et al., 2014*; *Burrows et al., 2019*; *Keel et al., 2022*; *Kerr et al., 2020*, *2022*). These data have provided tremendous insight into the spatial distribution of fish early life stages in this region, provided the first known spawning grounds for several taxa (*Kerr et al., 2020*), demonstrated a disparity between the composition of co-occurring egg and larval communities (*Burghart et al., 2014*), identified distinct distributions of eggs from neritic *vs.* oceanic taxa with a community transition at the shelf break (*Burrows et al., 2019*), and documented the potential of mesoscale cyclonic eddies to entrain the eggs of reef-associated taxa and transport them to deeper waters (*Kerr et al., 2020*). The goal of the SHELF project is to build a long-term time series of fish egg community composition on the West Florida Shelf at high spatial resolution to detect changes in fish-egg community composition over time; however, DNA barcoding of individual fish eggs is expensive and time-consuming. Two recent studies have demonstrated the use of metabarcoding, in which DNA is extracted and amplified from an aggregate sample containing all the fish eggs collected at a given site, to characterize the spawning community (*Duke & Burton, 2020*; *Miranda-Chumacero et al., 2020*). Here, we sought to evaluate the performance of metabarcoding as a potential way to increase throughput and reduce both financial and human resource costs to support a long-term fish egg monitoring program. We applied DNA metabarcoding to identify 4,719 fish eggs collected from 49 samples on the West Florida Shelf and recovered eggs from 37 taxa.

# MATERIALS AND METHODS

## Sample collection

We collected planktonic fish eggs from an *a priori*-defined grid on the West Florida Shelf (Fig. 1A) aboard the R/V *Hogarth* on two cruises (August 6–16, 2019, and September 24–26, 2019; Table S1). At each station, we performed a 15-min tow at the ocean surface with a bongo (double conical) plankton net (333 μm mesh, 61 cm mouth diameter)

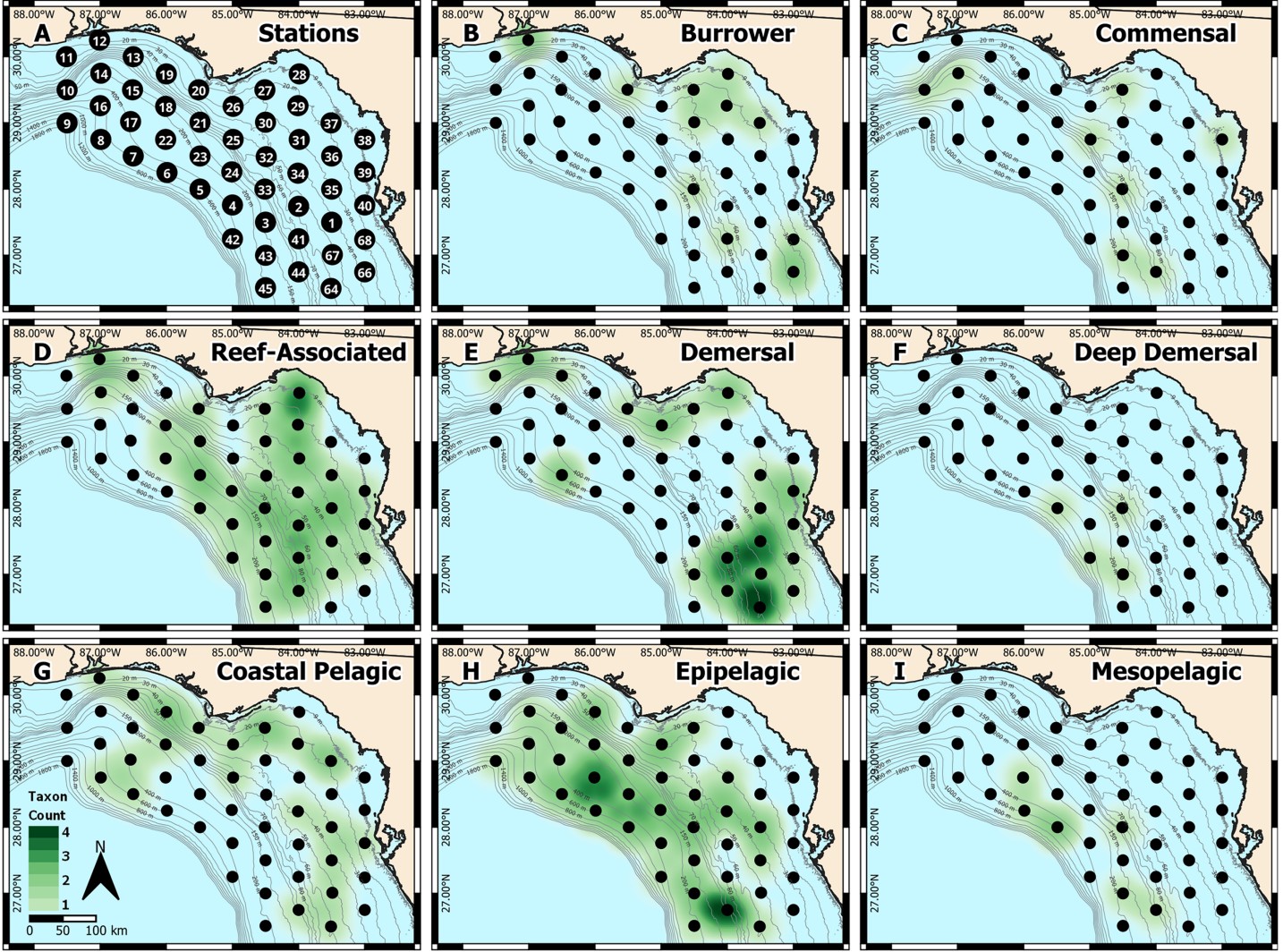

**Figure 1 West Florida Shelf study region and heatmap of identified taxa.** (A) Station locations were defined *a priori*, and heatmaps were based on the number of taxa identified at each station for each habitat type: (B) burrower, (C) commensal, (D) reef-associated, (E) demersal, (F) deep demersal, (G) coastal pelagic, (H) epipelagic, and (I) mesopelagic.

equipped with plastic, 1-liter cod-ends, and a General Oceanics 2030R mechanical flowmeter. After each tow, we washed down the nets with ambient seawater. The right-hand cod-ends were drained of excess seawater using a sieve and rinsed back into the jar using 95% isopropanol, leaving the final concentration >50% isopropanol. In the case of high biomass, we split the samples into two cod-ends to allow proper alcohol preservation. We stored the samples at 4 °C until processing.

In the laboratory, we picked at least 100 percomorph eggs (except when less were present) per sample using a stereomicroscope, gridded Petri dishes, and fine-tipped forceps during December 2019. When fewer than 100 eggs were present, we processed the entire sample. Each sample was separated into a labeled 1.5 mL screw cap, o-ring tube with 70% isopropanol for genetic identification. The number of eggs processed for each sample ranged from 2–272 (Table S1).

## DNA extraction, PCR, sequencing

To start the extraction process, we carefully removed the isopropanol with a sterile pipette tip. Next, we rinsed the eggs twice with molecular grade PCR water to remove any additional alcohol or other contaminants. To break open the chorion of the eggs, we added 0.4 $g$ of 1 mm beads to dry eggs along with 500 µl of HotSHOT alkaline lysis buffer (25 mM NaOH, 0.2 mM disodium EDTA, pH 12; *Truett et al., 2000*). We placed the tubes in a Fisher Scientific™ Bead Mill 4 Homogenizer for 5 min at 5 m/s and centrifuged briefly to reveal un-popped eggs. We manually broke any un-popped eggs with sterile toothpicks. We then incubated the tubes at 95 °C for 30 min, vortexing occasionally, and put them on ice for 3 min. Finally, we added 500 mL of HotSHOT neutralization buffer (40 mM Tris–HCl, pH 5; *Truett et al., 2000*) to each tube before storage at −20 °C until the Polymerase Chain Reaction (PCR) step.

We PCR amplified a 226 base pair (bp) region of the mitochondrial cytochrome c oxidase I (COI) gene using primer set Mini_SH-E, with forward primer 5′-CACGACGTTGTAAAACGACACYAAICAYAAAGAYATIGGCAC-3′, and reverse primer 5′-GGATAACAATTTCACACAGGCTTATRTTRTTTATICGIGGRAAIGC-3′ (*Shokralla et al., 2015*). Each 50-µl PCR contained final concentrations of 1× Apex $NH_4$ buffer, 1.5 mM Apex $MgCl_2$, 0.2 µM Apex dNTPs, 1 U Apex RedTaq (Genesee Scientific, San Diego, CA, USA), 0.2 µM forward and reverse primers, 10 µg/µl bovine serum albumin (New England BioLabs Inc., Ipswich, MA, USA), and 2 µl of target DNA (*Burrows et al., 2019*). The thermocycler conditions were as follows, 95 °C for 5 min, followed by 35 cycles of (94 °C for 40 s, 46 °C for 1 min, 72 °C for 30 s) and 72 °C for 5 min. We verified successful PCR amplification by running products on a 1.5% agarose gel stained with ethidium bromide. All samples were deemed successful and cleaned with the Zymo DNA Clean and Concentrator-25 Kit if the bands were bright, and the Zymo DNA Clean and Concentrator-5 Kit if the bands were faint. Negative controls, which did not contain eggs but underwent the entire extraction and amplification process, were processed alongside samples but never produced positive PCR products. We quantified the amplified DNA using a Qubit™ dsDNA HS Assay Kit, normalized the samples to equal concentrations, and sent to Genewiz for next-generation Illumina sequencing using the Genewiz Amplicon-EZ pipeline and partial Illumina adapters, forward 5′-ACACTCTTTCCCTACACGACGCTCTTCCGATCT-3′, reverse 5′-GACTGGAGTTCAGACGTGTGCTCTTCCGATCT-3′ to genetically identify the fish eggs present in each sample.

## Sequence analysis

To process the raw sequence data and obtain filtered and trimmed amplicon sequence variants (ASV), we used the Divisive Amplicon Denoising Algorithm (DADA2) v1.12 package (*Callahan et al., 2016*) in the *R* statistical environment (*R Core Team, 2022*). ASVs were first matched with species-level records in the Barcode of Life Database (BOLD; http://www.boldsystems.org/) (*Sujeevan & Hebert, 2007*), then BLASTn comparison (*Altschul et al., 1990*) against the National Center for Biotechnology Information (NCBI) nucleotide database (https://blast.ncbi.nlm.nih.gov/Blast.cgi) if no match was made in

BOLD. For some samples, more taxa were identified than the number of eggs present, signaling the presence of false positives and requiring the establishment of a threshold percentage of sequences required to consider a taxon present in a given sample (see "Discussion"). We applied a 2% threshold based on the total number of eggs within each sample and the number of taxa assigned. This empirically derived threshold is based on the principle that it would not be possible for a sample containing $n$ number of eggs to have greater than $n$ taxa present unless contamination were present. The final ASV table containing sequences that comprised >2% of the total sequence reads from any given sample is available in GRIIDC (*Kerr et al., 2023*).

### Quality control and data visualization

We identified sequences to the finest taxonomic resolution possible based on a comparison to BOLD. In some cases, we could not distinguish between multiple potential identifications based on the sequenced portion of the COI gene. Therefore, to refine our identifications, we referenced the geographic distribution of each taxon using published guides (*McEachran & Fechhelm, 1998*, *2005*) and FishBase (*Froese & Pauly, 2022*). We excluded any taxa not found in the Gulf of Mexico, such as those with Indo-Pacific or eastern Atlantic distributions. Twenty-one of the identifications were made at the species-level, while the remaining 16 identifications were to two or more closely related fishes; each distinct identification is referred to as a "taxon".

We also generated spatial heatmaps to visualize how metabarcoding-based identifications compared with the known habitat types occupied by each taxon (Fig. 1). Due to the qualitative nature of metabarcoding identifications, we used a presence-based approach. We first categorized each of the 37 distinct taxa into one of eight habitat types: burrower, commensal, reef-associated, demersal, deep demersal, coastal pelagic, epipelagic, and mesopelagic, based on information from FishBase (*Froese & Pauly, 2022*). We then quantified the number of distinct taxa in each habitat type at each sampling station. These presence-absence data were then used to generate heatmaps in QGIS (*QGIS Development Team, 2022*), where spatial weighting was set to 55 km (roughly equivalent to the distance between any two stations).

## RESULTS

We performed DNA metabarcoding on 4,719 fish eggs from 49 samples and obtained an average of 57,185 sequence reads per sample. The analyses presented here consider any taxa comprising >2% of the sequence reads from a given sample to be "present" in the sample and any sequences comprising <2% of the sequence reads from a given sample as "absent" since metabarcoding data are not quantitative due to methodological biases (see "Discussion"). We identified 37 distinct taxa, with 1–11 distinct taxa per sample (Table 1; Table S1). Twelve of the taxa (32%) were only detected in a single sample, eight taxa (22%) were detected in two samples, and the remainder were present in three or more samples. Tuna eggs (sequences could not be distinguished between *Auxis thazard/rochei*, *Euthynnus alletteratus*, and *Katsuwonus pelamis*) were identified in 26 samples, constituting the most widespread taxon in the dataset. Other taxa found at more than ten sites included

*Decapterus punctatus/tabl* (round/roughear scad), *Lutjanus griseus* (grey snapper), *Prionotus martis* (Gulf of Mexico barred searobin), *Pristipomoides aquilonaris* (wenchman), and *Xyrichtys sp.* (razorfish).

To validate the metabarcoding-based identification of fish eggs, we constructed spatial heatmaps based on presence-absence data (Figs. 1A–1I), which were consistent with the expected distributions of the identified taxa. Specifically, reef-associated species were found broadly throughout the sampling area (Fig. 1D), corresponding with the patchy distribution of structured hard-bottom throughout the West Florida Shelf (*Hine & Locker, 2011*). Coastal pelagic taxa were found inshore (Fig. 1G), epipelagic taxa were broadly distributed throughout the region (Fig. 1H), and mesopelagic taxa were found in deeper water farther offshore (Fig. 1I). Notably, we observed an apparent "hotspot" of demersal taxa (Fig. 1E) toward the southern end of the sampling region, which may indicate an area of interest for the management of recreationally or commercially important demersal fishes.

## DISCUSSION

DNA barcoding has gained popularity for identifying fish eggs; however, most studies analyze individual fish eggs. Processing individual fish eggs yields quantitative data; that is, we directly determine the exact proportion of the collected fish egg community comprised by each taxon. Information on the abundance of planktonic fish eggs from each taxon is valuable for estimating the biomass of parent fish stocks using the daily egg production method (*Stratoudakis et al., 2006*; *Burrows et al., 2019*). However, the number of eggs and sampling stations that can be processed with this method is limited by financial and labor resources. Metabarcoding of DNA extracted from all the collected fish eggs present at a given sample could be an advantageous alternative since it is faster and less expensive (*Cristescu, 2014*). If high spatial resolution of spawning sites is not required, numerous egg samples (*e.g.*, from a given season or oceanographic regime) could be pooled to further increase throughput and reduce costs. We are only aware of two studies to date that have applied metabarcoding to examine the community composition of fish eggs, one in marine waters and one in freshwater (*Duke & Burton, 2020*; *Miranda-Chumacero et al., 2020*, reviewed in *Lira et al., 2023*). Here we add to this emerging field of study by assessing the potential of metabarcoding as an alternate method for long-term monitoring of fish egg community composition. The advantages and disadvantages of DNA barcoding individual eggs *vs.* the metabarcoding method applied here are summarized in Table 2 and discussed below.

Compared to DNA barcoding of individual fish eggs, in which a single sequence is obtained from each egg, metabarcoding is not quantitative due to variable copy number of mitochondrial DNA in different taxa, different numbers of cells within eggs depending on developmental stage, chimeric sequences, and PCR amplification biases (*Bik et al., 2012*; *Hatzenbuhler et al., 2017*; *Duke & Burton, 2020*). A ground-truthing study by *Duke & Burton (2020)* demonstrated that fish egg metabarcoding reliably detected taxa that comprised over six percent of a mock community and three percent of a natural community, with variable recovery of rarer community members. Despite some variability,

**Table 1 Taxa comprising more than 2% of the sequences from any sample, habitat classification, and the number of stations where each taxon was identified.**

| Scientific name | Common name | Habitat | # Stations |
| --- | --- | --- | --- |
| *Acanthocybium solandri, Scomberomorus cavalla* | Wahoo/King Mackerel | Epipelagic | 3 |
| *Acanthostracion quadricornis* | Scrawled Cowfish | Reef | 6 |
| *Auxis thazard/rochei, Euthynnus alletteratus, Katsuwonus pelamis* | Bullet/Frigate Tuna, Little Tunny, Skipjack | Epipelagic | 26 |
| *Brama dussumieri/caribbea* | Lesser Bream/Carribean Pomfret | Deep demersal | 4 |
| *Callechelys muraena* | Blotched Snake Eel | Burrower | 1 |
| *Caranx crysos* | Blue Runner | Coastal pelagic | 5 |
| *Chaetodipterus faber* | Atlantic Spadefish | Reef | 1 |
| *Chilomycterus schoepfii/antillarum* | Striped/Web Burrfish | Reef | 1 |
| *Coryphaena hippurus* | Common Dolphinfish | Epipelagic | 2 |
| *Cyclopsetta fimbriata* | Spotfin Flounder | Demersal | 3 |
| *Decapterus punctatus/tabl* | Round/Roughear Scad | Coastal pelagic | 16 |
| *Diplogrammus pauciradiatus* | Spotted Dragonet | Reef | 1 |
| *Diplospinus multistriatus/Eustomias polyaster* | Striped Escolar/Dragonfish | Mesopelagic | 5 |
| *Echeneis naucrates/neucratoides, Remora remora/osteochir* | Live/Whitefin Sharksucker, Marlin/Sharksucker | Commensal | 8 |
| *Engraulis eurystole* | European/Silver Anchovy | Coastal pelagic | 1 |
| *Gordiichthys irretitus* | Horsehair Eel | Burrower | 1 |
| *Haemulon plumieri* | White Grunt | Reef | 1 |
| *Lepidopus altifrons* | Crested Scabbardfish | Mesopelagic | 1 |
| *Lutjanus campechanus* | Red Snapper | Reef | 2 |
| *Lutjanus griseus* | Grey Snapper | Reef | 11 |
| *Makaira nigricans* | Blue Marlin | Epipelagic | 2 |
| *Oxyporhamphus similis/micropterus* | Halfbeaks | Epipelagic | 4 |
| *Prionotus martis* | Gulf of Mexico Barred Searobin | Demersal | 11 |
| *Prionotus ophryas/scitulus* | Bandtail/Leopard Searobin | Demersal | 2 |
| *Prionotus roseus* | Bluespotted Searobin | Demersal | 2 |
| *Prionotus rubio/tribulus* | Blackwing/Bighead Searobin | Demersal | 8 |
| *Pristipomoides aquilonaris* | Wenchman | Reef | 18 |
| *Prognichthys occidentalis* | Bluntnose Flyingfish | Epipelagic | 2 |
| *Rachycentron canadum* | Cobia | Reef | 1 |
| *Rhomboplites aurorubens* | Vermilion Snapper | Reef | 2 |
| *Saurida normani/brasiliensis* | Shortjaw/Brazilian Lizardfish | Demersal | 2 |
| *Scomberomorus maculatus/regalis* | Atlantic Spanish Mackerel/Cero | Coastal pelagic | 1 |
| *Synagrops bellus/spinosus* | Blackmouth/Keelcheek Bass | Mesopelagic | 1 |
| *Synodus intermedius/foetens/ macrostigmus/sp* | Sand Diver/Inshore/Largespot/Lizardfish sp. | Demersal | 5 |
| *Thunnus atlanticus/albacares/sp* | Blackfin/Yellowfin/Tuna sp. | Epipelagic | 8 |
| *Trachinocephalus myops* | Snakefish | Demersal | 1 |
| *Xyrichtys novacula/sp* | Pearly Razorfish/Razorfish sp. | Burrower | 11 |

**Table 2 Comparison of individual egg DNA barcoding *vs.* metabarcoding.**

| Parameter | Individual eggs | Metabarcoding |
|---|---|---|
| Cost | $5.15 per egg<br>$494.40 per site[a] | $0.78 per egg[a]<br>$64.82 per site |
| Sequencing platform | Sanger | Illumina |
| Average sequence length | 500 base pairs | 200 base pairs |
| Quantitative | Yes | No |
| Ability to return to individual eggs with additional primers | Yes | No |
| Prevalence of false positives/negatives | Low/none | Frequent; dependent on the application of a threshold |

**Note:**
[a] Cost calculated based on 96 eggs per site.

*Duke & Burton (2020)* found a positive relationship between the proportion of reads from a given taxon and the proportion of eggs from that taxon in the mock communities. Therefore, although metabarcoding data cannot provide absolute taxon proportions, this technique can yield valuable information about abundant taxa and rarer taxa above a given threshold.

Another potential flaw with DNA metabarcoding of fish eggs is the detection of false positives either due to environmental DNA (eDNA) stuck on the fish egg surfaces or contamination introduced during processing (*Fritts et al., 2019*; *Duke & Burton, 2020*). *Duke & Burton (2020)* found that most false positives comprised a small percentage of the sequences recovered from a given sample. These data and other studies of fish early life stages suggest setting a threshold proportion of sequences required for counting a taxon as present; however, there is no consensus on what that threshold value should be, and it may need to be specific to each study area (*Mariac et al., 2018*; *Duke & Burton, 2020*; *Miranda-Chumacero et al., 2020*). In the present study, we empirically derived a threshold based on the total number of eggs within each sample and the number of taxa assigned, based on the principle that it would not be possible for a sample containing *n* number of eggs to have greater than *n* taxa unless contamination was present. We found that setting a 2% threshold (*i.e.*, sequences comprising less than 2% of the total sample reads were considered false positives and removed) ensured that the maximum number of taxa never exceeded the maximum number of eggs in a sample. If enough fish eggs are present, future work could perform direct comparisons between metabarcoding and individual egg barcoding by splitting samples and applying both methods to empirically determine the appropriate threshold for a given study region; however, our samples did not contain sufficient egg numbers to make this analysis possible.

Unlike individual egg DNA barcoding, where a single sequence is recovered from each egg and the results are quantitative, we found that the metabarcoding results varied depending on the threshold applied; thus false positives (*i.e.*, sequences above the threshold that were not derived from eggs) and false negatives (*i.e.*, sequences below the threshold that were derived from eggs) remain problematic and can have a major effect on reported spawning sites. For example, the 2% threshold used to analyze the data in this study resulted in an average of 3.67 taxa per sample (range 1–11). Applying a more conservative

5% threshold would have resulted in an average of 2.18 taxa per sample (range 1–6). Although 32 of the 37 taxa identified in this study would still be detected in at least one sample with the 5% threshold, five taxa (*Acanthocybium solandri/Scomberomorus cavalla*, *Engraulis eurystole*, *Haemulon plumieri*, *Scomberomorus maculatus/regalis*, *Synagrops bellus/spinosus*) would have been removed completely. Eggs from all five of these taxa have been previously recovered on the West Florida Shelf through individual egg barcoding (*Keel et al., 2022*; *Kerr et al., 2022*), so in this case, we believe that increasing the threshold would likely result in false negatives. This example demonstrates the large effect that small differences in the threshold can have on DNA metabarcoding results.

Since the inception of the Fish Barcode of Life (FISH-BOL) Initiative, the COI gene is widely used for genetic identification of fishes and this gene is capable of distinguishing between the majority of described fish species (*Teletchea, 2009*; *Ward, Hanner & Hebert, 2009*). The *Shokralla et al. (2015)* primers applied in this study were validated on 8,000 DNA barcodes from North American fish species and have been widely adopted by the community. However, these primers were not able to resolve all of the taxa identified in this study to species level, suggesting that further optimization could improve taxonomic resolution. Numerous studies have demonstrated the advantage of using multiple genetic markers for metabarcoding (*Evans et al., 2016*; *Sawaya et al., 2019*; *Duke & Burton, 2020*); however, databases tend to be more limited for other markers and may need to be supplemented for geographic regions of interest. One advantage of individual egg barcoding is the ability to return to DNA samples extracted from specific eggs to analyze population genetics or apply additional primer sets in the case where the conserved region of the COI gene used for barcoding cannot distinguish between certain species complexes. For example, *Burrows et al. (2019)* applied additional PCR primers to distinguish between the economically important species *Thunnus thynnus* and *Katsuwonus pelamis*, as well as between *Scomberomorus cavalla* and *Acanthocybium solandri*, to achieve a definitive identification. The ability to return to specific eggs to refine taxonomic uncertainties is lost in DNA metabarcoding, where the DNA from all eggs within a sample is combined. Finally, it is possible that the shorter sequence length used for Illumina sequencing in metabarcoding compared to Sanger sequencing used for individual egg barcoding might hinder the assignment of sequences to species level. We did not experience lower taxonomic resolution due to the shorter sequence length obtained in this study compared to our prior work with longer COI sequences, which is consistent with other analyses that have shown the ability to reliably assign 140 bp reads with relatively high success rates (*Shokralla et al., 2015*; *Kimmerling et al., 2018*; *Mariac et al., 2018*).

PCR biases represent an important barrier to the feasibility of metabarcoding studies for quantitative analyses of fish early life stages (*Lamb et al., 2019*; *Zinger et al., 2019*). Efforts to make metabarcoding of fish larvae more quantitative have shown success, although these methods have not yet been applied to eggs and have only been examined in a limited number of studies and regions. Applying a different approach to quantify fish early life stages, *Kimmerling et al. (2018)* used high coverage metagenomic sequencing (sequencing of total DNA from a given sample, without first applying PCR to enrich for the COI gene). *Kimmerling et al. (2018)* adjusted the sequence coverage for each sample to obtain ~20

COI-derived reads per larva, allowing samples with more larvae to be sequenced more deeply. These methods yielded quantitative results when normalized by the relative size of each larva in the sample, showing the promise of this technique (*Kimmerling et al., 2018*). However, this method requires a large amount of sequencing since the percentage of metagenomic sequence reads that belonged to the COI gene was extremely low (approximately 1 in every 18,000 sequences), which will present a substantial barrier for long-term monitoring efforts. With a newly developed Metabarcoding by Capture using a Single Probe (MCSP) method, *Mariac et al. (2018)* achieved ~6,000 times enrichment of COI sequences compared to an unenriched sample. By analyzing a mock community, the relative frequencies of sequences recovered from larval swarms in the Amazon basin with the MCSP method correlated well with true frequencies derived from Sanger sequencing of individual fish larvae (*Mariac et al., 2018*). Since this method relies on hybridization instead of PCR amplification, MCSP is subject to fewer biases. However, it should be noted that MCSP still required the application of a threshold value for the minimum number of reads per taxon in order to count that taxon as present. Even with the application of an empirically defined threshold established through the analysis of mock communities, a small number of false positives were still encountered (*Mariac et al., 2018*).

Unfortunately, we cannot directly compare the metabarcoding results presented here to our prior surveys of individual fish eggs in the GOM since the samples were collected in different years and locations, and often with different sampling methods. More taxa were identified in individual egg barcoding studies (*e.g.*, *Burrows et al., 2019*) due to the ability to confidently identify every single egg without needing to apply a threshold. Therefore, for the purposes of the SHELF project, individual egg barcoding definitively identifies spawning sites for more taxa. Often only one or two eggs are identified from a given species at each site and many species are only encountered at a single site. However, it is notable that the vast majority of the taxa identified here have also been recovered from this region in our prior work (*Keel et al., 2022*; *Kerr et al., 2022*). The only exceptions found with metabarcoding that we have not observed with our more spatially and temporally expansive individual egg barcoding were *Makaira nigricans* (Atlantic blue marlin), *Lepidopus altifrons* (crested scabbardfish), and *Chilomycterus sp.* (burrfish), all of which are known to occur in the GOM. We examined the data for these taxa to determine if they were present in very low abundances and thus were likely false positives; however, that was not the case. *Chilomycterus* sp. comprised ~87% of the sequences in one sample, *Lepidopus altifrons* comprised ~20% of the sequences in another sample, and *Makaira nigricans* was found in two samples, where it made up ~16% of the sequences of each sample. This suggests that eggs from these taxa were truly present in these samples, and demonstrates an advantage of metabarcoding; namely, by enabling the processing of samples from more stations, we can capture rarer or more episodic spawning events.

## CONCLUSIONS

In this study, we assessed the performance of DNA metabarcoding to increase throughput and reduce financial and labor costs associated with a long-term fish egg monitoring program. A total of 37 taxa were identified from 49 stations on the West Florida Shelf.

Egg identifications were consistent with prior species distributions observed from individual egg DNA barcoding and spatial heatmaps of eggs corresponded to known habitat types occupied by adults. The increased throughput allowed by metabarcoding resulted in the identification of taxa not previously detected in this region, possibly representing episodic spawning events. One disadvantage of metabarcoding is that this method is not quantitative and requires the application of a threshold proportion of sequences required to count a taxon as present. The choice of DNA barcoding methods therefore depends on the goals of the study, and fisheries monitoring efforts may benefit from a combination of the two approaches, with individual egg barcoding providing quantitative information and metabarcoding expanding the number or geographic range of samples that can be processed.

## ACKNOWLEDGEMENTS

The authors thank the volunteers that collected plankton samples and the captain and crew on the R/V Hogarth in August and September, 2019. Thanks to Claudia Baron-Aguilar, Brianna Michaud, Vi Nguyen, Julie Vecchio, and Amy Wallace for help picking fish eggs from the plankton samples, and to Natalie Sawaya for guidance with bioinformatics.

### Funding

This research was made possible by research grants from the Florida Institute of Oceanography's RESTORE Act Centers of Excellence Program (USF Projects 4710112601 and 4710112901). The funders had no role in study design, data collection and analysis, decision to publish, or preparation of the manuscript.

### Grant Disclosures

The following grant information was disclosed by the authors:
Florida Institute of Oceanography's RESTORE Act Centers of Excellence Program (USF Projects): 4710112601 and 4710112901.

### Competing Interests

Mya Breitbart is an Academic Editor for PeerJ.

### Author Contributions

- Mya Breitbart conceived and designed the experiments, performed the experiments, analyzed the data, prepared figures and/or tables, authored or reviewed drafts of the article, and approved the final draft.
- Makenzie Kerr performed the experiments, analyzed the data, prepared figures and/or tables, authored or reviewed drafts of the article, and approved the final draft.
- Michael J. Schram analyzed the data, prepared figures and/or tables, authored or reviewed drafts of the article, and approved the final draft.
- Ian Williams performed the experiments, analyzed the data, prepared figures and/or tables, authored or reviewed drafts of the article, and approved the final draft.

- Grace Koziol performed the experiments, analyzed the data, prepared figures and/or tables, authored or reviewed drafts of the article, and approved the final draft.
- Ernst Peebles conceived and designed the experiments, performed the experiments, authored or reviewed drafts of the article, and approved the final draft.
- Christopher D. Stallings conceived and designed the experiments, authored or reviewed drafts of the article, and approved the final draft.

## Data Availability

The data are available in GRIIDC: Kerr, Makenzie, Grace Koziol, Ernst B. Peebles, Chris Stallings, and Mya Breitbart. 2023. Identification with metabarcoding of fish eggs collected aboard the R/V Hogarth from 2019-08-06 to 2019-09-26 on the west Florida shelf. Distributed by: Gulf of Mexico Research Initiative Information and Data Cooperative (GRIIDC), Harte Research Institute, Texas A&M University–*Corpus* Christi. DOI 10.7266/7pvp5zc1.

## Supplemental Information

Supplemental information for this article can be found online at http://dx.doi.org/10.7717/peerj.15016#supplemental-information.

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
