# Peer review of "Evaluation of DNA metabarcoding for identifying fish eggs: a case study on the West Florida Shelf"

_PeerJ, doi:10.7717/peerj.15016_

## Round 0.1 · original submission · Minor Revisions

I've had this manuscript evaluated by 3 reviewers, and all have provided a number of comments that I think will improve the paper. Reviewer 1 in particular had a few concerns about primer reliability, sample sizes, etc -- and addressing or commenting on those in the revision would be important.

Reviewer 1 ·

Basic reporting

This manuscript is about the application of metabarcoding to analyze the fish eggs for the scientific management and conservation of fish genetic resources in Florida. Recently, genomic approach for the ecological survey are being adopted and this is one of the good trial, which will replace time and labor-consuming traditional methodology with fast and more feasible modern technology. The most important aspect of this trial is to increase sample size and numbers composing a big data set to provide more accurate and statistically proven data for the scientific management of fish resources. Therefore, I believe application of metabarcoding for fish egg survey would be one of eminent task to resolve now in fishery and oceanographic studies.
Despite the importance of this topic, I am afraid that authors in this study may misunderstand the application of metabarcoding, potentially misleading readers. I found that average obtained fish species from each index were approximately 4-5 at most using metabarcoding, which is real waste of resources. Collectively, they obtained only 47 species among which 32 % were detected only once in this survey. Metabarcoding analyze can analyze total target taxa from the mixed sample with low cost and labors. I would rather sequence all the pooled sample with single index and increase sample collection numbers with different season or oceanographic characteristics. If we would like to indentify only for species composition after manual selection of eggs from the net samples, this method may not be suitable. As authors also discussed in the manuscript, one-time read to identify the species may not be suitable using metabarcoding. For instance, with different time there would be some patterns of each species according to season or oceanographic condition, or etc. However, In this study, authors tested only one time survey and compare the result with traditional microscopic observation. I believe two methods are totally different object. Metabarcoding would provide a more statistical numbers rather than qualitative data, while microscopic observation can also explain more than just species, including developmental states, morphological characteristics, or more. Therefore, I think this single run may not explain any of characteristics of metabarcoding analysis.
Besides, authors did not conducted test to know the reliability of primer used. In order to know the reliability of the primer used, authors should have a conducted either in-silico or mock sample analysis to convince readers the reliability of universal primer used for fish taxa. I also strongly suggest that authors should have conducted an environmental DNA survey to obtain regional fish haplotype reference dataset. I failed to find any explanation of haplotypes obtained in this study and how each haplotypes can be assigned in each species. I think not all obtained ASVs match 100 % to the species name if there has not been any previous survey in the florida area. I was also surprised to know that authors arbitrarily eliminate the obtained fish reads by empirical judgement. According to field eDNA sample guide line (https://www.csagroup.org/ article/research/environmental-dna-standardization-needs-for-fish-and-wildlife-population-assessments-and-monitoring), lab facilities should be ready to eliminate the cross-contamination and negative- and positive- control should be used throughout the experiment. We should know that copy numbers of the unfertilized egg may be lower than the cross contaminated DNAs while handling. Therefore, I believe this experiment is nothing but a preliminary test for metabarcoding analysis for fish eggs in the Florida water. I suggest that they should validate the reliability of primer used, reference dataset, and establishment of experimental procedures. For further review consideration, I would like to see some additional analyses with different season

Experimental design

As explain above

Validity of the findings

As explained above

Additional comments

N/A

Reviewer 2 ·

Basic reporting

The paper is well written and meets all these standards.

Experimental design

The question addressed and methods are well defined.

Validity of the findings

Data are provided and conclusions are well stated.

Additional comments

Over all, this is a well-written and interesting study addressing an important but largely unaddressed issue. Metabarcoding is certainly an attractive approach to monitoring and seems to be gaining traction as the method of choice, but we really need more studies that compare this approach to other methods, such as analysis of individual eggs. This paper does a nice job giving the general pros and cons of the approach. I agree with the conclusion that choice of method will depend on the goals of the study.

It would have been great if direct comparisons of barcoding and metabarcoding could be made. Given previous work using single egg barcoding, the authors should be more explicit why direct comparisons of the two approaches aren’t possible. Don’t the sampling areas overlap and can’t an analysis be done only in the region of overlap? Such an analysis would make the study more powerful. Without some direct comparison to single egg identifications, how do we know what species metabarcoding might be missing? Lines 295-306 point to cases where metabarcoding found three species not picked up in previous single egg studies – were there any species found in single egg studies that were not found in the metabarcoding data? I was also surprised that metabarcoding found low species diversity in most samples - does this align well with previous single egg barcoding studies?

In the discussion, the authors do a good job noting the advantages of single egg barcoding in terms of quantitative data and lack of false positives. They don’t say much about the fact that single egg PCR uses longer COI barcodes and therefore sometimes does a better job of resolving species. The data presented are remarkable in that of 37 different “taxa” observed, only 21 were resolved to species level. Some of this is certainly expected – the Thunnus sp are 99% identical at COI. But Katsuwonus pelamis / Euthynnus alletteratus / Auxis are only ~92% identical and certainly commercially important. Seems like you'd want to use an amplicon that distinguishes these species? So maybe expand upon the choice of the amplicon used in the study.

Error in line 163 -- 55,000 km ?? This is greater than the circumference of the earth.

·

Basic reporting

English is acceptable, but minor adjustments can be made to improve your text. See notes in pdf. See the paper of Lira et al. 2022; there is a table showing references of articles with metabarcoding of fish eggs and larvae; I think the paper will also be helpful.

Experimental design

No comment.

Validity of the findings

No comment.

---

## Round 0.2 · accepted · Accept

The revision appears to have addressed all of the comments of the reviewers, congratulations on the publication!